# Fairness-aware Prompt Tuning for Graph Neural Networks

Submission Id: 2856*

## Abstract

Graph prompt tuning has achieved significant success for its ability to effectively adapt pre-trained graph neural networks to various downstream tasks. However, the pre-trained models may learn discriminatory representation due to the inherent prejudice in graph-structured data. Existing graph prompt tuning overlooks such unfairness, leading to biased outputs towards certain demographic groups determined by sensitive attributes such as gender, age, and political ideology. To overcome this limitation, we propose a fairness-aware graph prompt tuning method to promote fairness while enhancing the generality of any pre-trained GNNs (named FPrompt). FPrompt introduces hybrid graph prompts to augment counterfactual data while aligning the pre-training and downstream tasks. It also applies edge modification to increase sensitivity heterophily. We provide a two-fold theoretical analysis: first, we demonstrate that FPrompt possesses universal capabilities in handling pre-trained GNN models across various pre-training strategies, ensuring its adaptability in different scenarios. Second, we show that FPrompt effectively reduces the upper bound of generalized statistical parity, thereby mitigating the bias of pre-trained models. Extensive experiments demonstrate that FPrompt outperforms baseline models in both accuracy and fairness (~33%) on benchmark datasets. Additionally, we introduce a new benchmark for transferable evaluation, showing that FPrompt achieves state-of-the-art generalization performance.

## 1 Introduction

Graph Neural Networks (GNNs) have been successfully applied across a wide range of domains, including social network analysis [5], anomaly detection [11, 12], and recommendation systems [16]. However, traditional GNNs often rely on large amounts of labeled data, which can be scarce in real-world applications [43]. Furthermore, these models frequently struggle with poor generalization when faced with out-of-distribution data [34]. To address these limitations, researchers have increasingly explored pre-training and fine-tuning strategies for GNNs [22], inspired by the success of similar approaches in natural language processing [24]. In this paradigm, a GNN is pre-trained on a massive corpus of graph datasets, then fine-tuned for specific downstream tasks, leveraging the knowledge acquired during pre-training.

While pre-training and fine-tuning on graphs have shown promising results, there are still several challenges. One significant issue is the gap between pre-training objectives, such as edge prediction in self-supervised tasks [14] and the goals of downstream tasks like node or graph classification [22]. This misalignment often leads to sub-optimal performance during fine-tuning [19]. Additionally, pre-trained models are prone to catastrophic forgetting when adapted to new tasks, where the model forgets previously learned knowledge during the fine-tuning process [42].

In response to these challenges, prompt tuning has recently emerged as a compelling alternative for tuning pre-trained GNNs [9, 20, 27, 38]. Rather than fine-tuning model parameters, prompt tuning modifies the input data to better align with the downstream task, leaving the parameters of the pre-trained GNNs unchanged. For instance, GraphPrompt [20] pre-trains GNNs on link prediction tasks and adapts downstream node or graph classification tasks to this pre-training task format by introducing prompt-specific parameters such as class prototypes. This approach allows for efficient adaptation while preserving the integrity of the pre-trained model.

Despite its success, current graph prompt tuning techniques neglect the issue of bias present in pre-trained models. Bias can arise from the graph-structured data used during pre-training, as numerous studies have demonstrated that historical data often contains patterns of discrimination related to sensitive attributes like age, gender, race, and region [6, 23, 33]. Additionally, commonly used pre-trained GNN models such as GCN [17] and GAT [31] do not inherently address fairness, and their message-passing mechanisms may even amplify existing biases [26]. Our experiments in Table 2 further indicate that existing graph prompt tuning tends to exacerbate biases in pre-trained models. This largely limits the real-world applicability of graph prompt tuning in fairness-aware domains.

Therefore, a natural question is raised: *can we develop fairness-aware graph prompt tuning that retains the benefits of efficient adaptation while ensuring fairness?* Nevertheless, answering the above question is technically challenging: (**i**) Existing fairness approaches are often tailored to specific datasets, while in practice, the same pre-trained GNNs are expected to be adapted to various datasets without the need for further parameter tuning. Therefore, it is highly non-trivial to directly apply existing fairness methods to eliminate bias in pre-trained GNNs; and (**ii**) most existing fairness methods lack theoretical analysis [29], meaning they do not provide a practical guarantee, i.e., provable upper bounds on common fairness metrics such as statistical parity.

To address these issues, in this paper, we propose a fairness-aware graph prompt tuning method named FPrompt, that could alleviate bias in any pre-trained GNNs and enhance their adaptability at the same time. Specifically, FPrompt introduces hybrid graph prompts, which include both fixed and learnable prompts. The fixed prompts are designed to represent sensitive group embeddings, and their interaction with the original graph can be viewed as counterfactual data augmentation to mitigate bias. The learnable graph prompts, which adopt the token as graph prompt, effectively adapt the pre-trained model to different datasets. Additionally, FPrompt predicts the sensitive attributes of nodes and applies heterophily-enhanced edge modification based on the assignments. This process enhances information flow between nodes in different sensitive groups, thereby reducing the representational disparity across these groups. We provide a two-part theoretical analysis: (i) we establish that FPrompt exhibits a high degree of versatility in handling pre-trained GNN models, regardless of the specific pre-training strategy employed. This flexibility ensures that FPrompt can be seamlessly

integrated into a wide range of applications and use cases, adapting to different scenarios with ease; and (ii) we demonstrate that FPrompt can reduce the upper bound of generalized statistical parity, effectively addressing and mitigating the inherent biases present in pre-trained models. To validate the cross-dataset fairness performance of different fine-tuning strategies, we construct a new benchmark consisting of four real-world datasets. These datasets share the same feature dimensions, which eliminates the potential fairness impact caused by traditional cross-dataset generalization experiments involving singular value decomposition operations. In summary, our main contributions are as follows:

- To the best of our knowledge, we are the first to propose a fairness-aware graph prompt tuning method that can mitigate bias in any pre-trained models and enhance their generalization ability at the same time;
- We provide theoretical analysis of standard fairness metrics for the pre-training and fine-tuning paradigm, establishing a solid theoretical foundation for FPrompt's performance; and
- We construct a new benchmark to evaluate the cross-dataset fairness performance of fine-tuning strategies. Extensive experiments on different scenarios demonstrate that FPrompt achieves state-of-the-art performance in both accuracy and fairness.

## 2 Related Work

### 2.1 Graph Neural Networks

Graph Neural Networks (GNNs) have become the predominant framework for learning on graph-structured data. They can be broadly categorized as spectral- and spatial-based GNNs. Spectral-based GNNs rely on the graph Fourier transform, which operates in the spectral domain of the graph Laplacian. The pioneering work by Bruna et al. [2] introduced the concept of using graph convolution in the spectral domain. This was later simplified in GCN [17], which approximates the spectral convolution operation using 1-order Chebyshev polynomials to improve computational efficiency. However, spectral methods typically depend on the eigen-decomposition of the graph Laplacian, making them less scalable and limited to transductive learning. In contrast, spatial-based GNNs define convolutions in the graph's spatial domain by aggregating features from a node's local neighborhood [8, 14, 31, 37]. These models support inductive learning and offer better scalability to large-scale graphs.

Traditional GNNs are typically trained in a supervised manner, requiring a substantial amount of labeled data. However, this reliance on labeled data presents challenges in real-world applications, where labels may be sparse or costly to obtain, and models often face poor transferability to new domains. Recently, unsupervised graph representation learning has gained a lot of attraction. Early attempts like DeepWalk [25] and node2vec [13], employ random walks to transform graph learning into a sequence learning problem. While these methods are scalable, they largely focus on local neighborhoods, overlooking broader graph structures. To overcome these limitations, more recent approaches [32, 36, 39, 40] apply contrastive learning techniques to maximize mutual information between local and global node representations. This allows the models to generate more meaningful, transferable embeddings without labeled data, achieving competitive performance on tasks such as

node classification and graph classification [3]. For example, GCL designs four types of graph augmentations to incorporate various priors [40]. GRACE takes the original graph as input and GNN model with its perturbed version as two encoders to obtain two correlated views for contrast [35].

### 2.2 Graph Prompt Tuning

Graph prompt tuning has gained significant attention due to its effectiveness in bridging the gap between pre-training and downstream objectives. Due to its parameter-efficient nature, graph prompt tuning has quickly become popular as an alternative to fine-tuning large pre-trained models, particularly in scenarios with limited downstream task labels [38]. For instance, GraphPrompt [20] presents a unified framework that relies on subgraph similarity and link prediction, utilizing a learnable prompt to guide downstream tasks by incorporating task-specific aggregation in the readout function. Additionally, it computes class prototype vectors through supervised prototypical contrastive learning. GPF [9] extends node embeddings by integrating task-specific prompt parameters, making it adaptable to any pre-trained GNN model, regardless of the pre-training strategy. All-in-one [28] reformulates node-level and edge-level tasks into graph-level tasks and introduces meta-learning techniques into graph prompt tuning. For a more comprehensive summary of graph prompt tuning methods, we refer readers to [29]. However, these approaches largely overlook the inherent bias in pre-trained models, which significantly limits the real-world applicability of graph prompt tuning in fairness-sensitive domains.

### 2.3 Fairness in Graph Representation Learning

Recently, with the rapid advancements and widespread adoption of Graph Neural Networks (GNNs), concerns about fairness within these models are attracting increasing attention. Algorithmic fairness in GNNs can be broadly categorized into two main types: individual fairness and group fairness. On the one hand, individual fairness requires that similar individuals (or nodes) in the graph receive similar treatment or outcomes. On the other hand, group fairness focuses on ensuring that specific disadvantaged or protected groups are not unfairly treated in comparison to other groups, addressing concerns like bias against minorities or marginalized communities. For example, FairGNN [4] enhances fairness by reducing the identifiability of sensitive attributes within node embeddings through adversarial training. FairVGNN [33] introduces a feature masking strategy to prevent sensitive information leakage during the feature propagation process in GNNs. NIFTY [1] aims to maximize the agreement between the original graph and its counterfactual augmented views to promote fairness. However, these fairness-aware models require optimizing a fair GNN for each specific dataset and cannot be directly applied to mitigate bias in pre-trained models, where a single pre-trained model is expected to be adapted across different downstream datasets.

## 3 Preliminaries

### 3.1 Pre-training, Prompt, and Fine-tuning

We use $\mathcal{G} = (\mathbf{A}, \mathbf{X})$ to denote a graph, where $\mathcal{V}$ and $\mathcal{E}$ denotes the node and edge sets, respectively. $\mathbf{A} \in \mathbb{R}^{N \times N}$ is the adjacency matrix, where $N$ is the number of nodes. $\mathbf{X} \in \mathbb{R}^{N \times F}$ is the node feature

matrix, where each node $v_i$ is associated with a $F$-dimensional attribute vector $\mathbf{x}_i \in \mathbb{R}^F$. Furthermore, each node $v_i$ has a binary sensitive attribute $s_i \in \{0, 1\}$. The sensitive group is defined as $\mathcal{S}_k = \{v_i | s_i = k\}, k = \{0, 1\}$.

In general, the pre-training, prompting and fine-tuning paradigm of a graph model consists of three main components: a GNN backbone $\Psi : (\mathbf{X}, \mathbf{A}) \rightarrow \mathbf{H} \in \mathbb{R}^{N \times P}$, a prompting function $\Phi : (\mathbf{X}, \mathbf{A}) \rightarrow (\mathbf{X}', \mathbf{A}')$, and an adapter $\Xi : \mathbf{H} \rightarrow \mathbf{Z} \in \mathbb{R}^{N \times C}$. In the pre-training phase, the goal is to optimize the parameters of the GNN backbone $\Psi$ through self-supervised learning, enabling it to capture useful information from the graph structure [35, 40]. In the prompting phase, the original graph is transformed into a prompted graph via the prompting function $\Phi$. Commonly used graph transformations include generating the new feature matrix, adding or removing edges, and adding or removing sub-graphs [9, 28]. In the fine-tuning phase, the parameters of the GNN backbone are frozen, and only the parameters in the prompting function $\Phi$ and the adapter $\Xi$ need to be optimized for different downstream tasks. In this paper, we consider the node classification task where each node $v_i$ belongs to only one class $y_i$. After pre-training, prompting, and fine-tuning, the output can be written as $\mathbf{Z} = \Xi \circ \Psi \circ \Phi(\mathbf{X}, \mathbf{A}) \in \mathbb{R}^{N \times C}$, where $C$ is the class number. The predicted label $\hat{y}_i$ of node $v_i$ can be obtained from the $i$-th row $\mathbf{z}_i$ of $\mathbf{Z}$.

## 3.2 Fairness Measurement

Fairness ensures that no individual or group faces unjust treatment based on sensitive attributes such as race and gender. Our paper focuses on group fairness [4, 33], which asserts that a model's outcomes should treat groups with differing sensitive attributes equitably. Nonetheless, in many instances, models may exhibit a bias that disproportionately benefits one group over another, leading to unfair outcomes [6]. Various fairness metrics have been proposed to evaluate how models perform across different demographic groups. These metrics are commonly formulated within the context of binary classification, where $y_i \in \{0, 1\}$.

*Definition 3.1 (Statistical Parity [7]).* Statistical parity requires the predictions to be independent with the sensitive attribute $s$, i.e.,

$$\mathbb{P}(\hat{y}_i \mid s_i = 0) = \mathbb{P}(\hat{y}_i \mid s_i = 1). \tag{1}$$

*Definition 3.2 (Equal Opportunity [15]).* Equal opportunity requires the probability of an instance in a positive class being assigned to a positive outcome should be equal for both subgroup members. The property of equal opportunity is defined as:

$$\mathbb{P}(\hat{y}_i = 1 \mid y_i = 1, s_i = 0) = \mathbb{P}(\hat{y}_i = 1 \mid y_i = 1, s_i = 1). \tag{2}$$

Following [4], we apply the following metrics to quantitatively evaluate statistical parity and equal opportunity:

$$\Delta_{SP} = |\mathbb{P}(\hat{y}_i = 1 \mid s_i = 0) - \mathbb{P}(\hat{y}_i = 1 \mid s_i = 1)|,$$
$$\Delta_{EO} = |\mathbb{P}(\hat{y}_i = 1 \mid y_i = 1, s_i = 0) - \mathbb{P}(\hat{y}_i = 1 \mid y_i = 1, s_i = 1)|, \tag{3}$$

where the probabilities are evaluated on the test set. For both metrics, smaller values indicate better fairness. Finally, we define the node-level fairness homophily.

*Definition 3.3 (Fairness Homophily [33]).* For node $v_i$ in a graph $\mathcal{G}$, we define its fairness homophily ratio $h_i$ as:

$$h_i = \frac{| \left\{ (v_i, v_j) : v_j \in \mathcal{N}_i \wedge s_i = s_j \right\} |}{|\mathcal{N}_i|}. \tag{4}$$

The fairness homophily of the graph is defined as $h_{\mathcal{G}} = \sum_{i=1}^{N} h_i / N$.

It is important to note that, unlike the traditional definition of homophily [3], where homophily is typically defined based on node labels $y_i$, in this context, $h_i$ depends on the sensitive attribute $s_i$. This means that instead of measuring how nodes with the same labels tend to connect, we focus on how nodes with similar sensitive attributes (such as gender) tend to interact with each other.

## 3.3 Problem Formulation

Following existing models [4, 6], we focus on the binary class and binary sensitive attribute setting, i.e., both $y_i$ and $s_i$ can either be 0 or 1 for each node $v_i$. Given a subset of nodes with labels $\mathcal{V}^L$ and sensitive attributes $\mathcal{S}^L$, our goal is to design a tuning method that can be applied to any pre-trained GNNs without modifying the pre-trained model's parameters. The output of the pre-trained and fine-tuning graph model should maintain high accuracy while satisfying the fairness criteria such as statistical parity.

## 4 Methodology

In this section, we propose a fairness-aware prompt tuning method for GNNs named FPrompt. The key ingredient of FPrompt is a novel prompting function that applies graph transformation to both the features and adjacency matrix:

$$\Phi^{(FPrompt)} : (\mathbf{X}, \mathbf{A}) \rightarrow \left( \mathbf{X}^{(FPrompt)}, \mathbf{A}^{(FPrompt)} \right). \tag{5}$$

where $\mathbf{X}^{(FPrompt)} \in \mathbb{R}^{N \times F}, \mathbf{A}^{(FPrompt)} \in \mathbb{R}^{N \times N}$ represent the transformed graph features and adjacency matrix, respectively. To this end, the prompting function consists of two essential components: (i) *hybrid graph prompts* that promote fairness through generating counterfactual features $\mathbf{X}^{(FPrompt)}$, while narrowing the gap between pre-training and downstream tasks via learnable tokens (Section 4.1); (ii) *heterophily-enhanced edge modification* that mitigates bias by modifying the graph structure to $\mathbf{A}^{(FPrompt)}$ to increasing message passing between different sensitive groups (Section 4.2). Through these designs, FPrompt can alleviate bias in any pre-trained GNNs by merely fine-tuning the parameters in the prompt function and the adapter without altering the parameters of the pre-trained models. We introduce the detailed fine-tuning strategy of FPrompt in Section 4.3. We summarize the framework of FPrompt in 1.

## 4.1 Hybrid Graph Prompts

Graph prompts are designed to bridge the gap between the pre-training task and the downstream task. They allow for efficient learning by adapting the model through learnable prompts rather than full-scale retraining, achieving high-quality outcomes while keeping computational costs low. However, existing graph prompts overlook potential biases in pre-trained graph models, leading to unfair outputs. To address this issue, we propose hybrid graph prompts that consist of fixed graph prompts and learnable graph prompts.

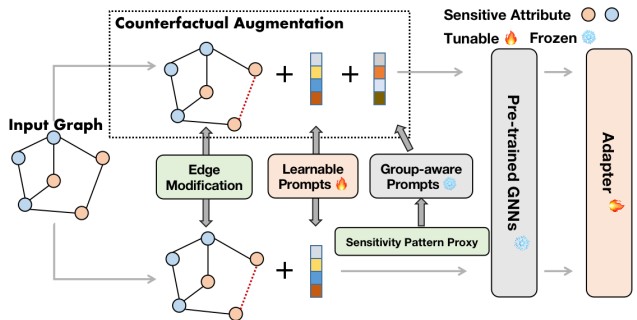

**Figure 1: The framework of FPrompt.**

Both types of graph prompts are added to the original features to generate new features $\mathbf{X}^{(FPrompt)}$. The fixed graph prompts are designed to enhance fair representation learning through counterfactual data augmentation. Meanwhile, the learnable graph prompts aim to narrow the gap between pre-training and downstream tasks (*e.g.* node classification).

*4.1.1 Fixed graph prompts.* The purpose of fixed graph prompts is to create counterfactual features. Specifically, the fixed graph prompts contain two sensitivity-group-aware prompt tokens $\mathcal{P} = \{p_0, p_1\}$, where each token $p_s \in \mathcal{P}$ can be represented by a token vector $\mathbf{p}_s \in \mathbb{R}^F$. We obtain $\mathbf{p}_s$ by calculating the average feature of the nodes with sensitive attribute $s$ as

$$\mathbf{p}_s = \text{mean}(\mathbf{x}_i \mid v_i \in \mathcal{S}_k^L) \tag{6}$$

The token $\mathbf{p}_s$ captures the general feature patterns of nodes within the sensitive group $\mathcal{S}_s$, serving as a proxy of the group's characteristics. By incorporating these tokens, we can infuse information from different sensitive groups, allowing for the generation of counterfactual representations that promote fairness. Next, we obtain the generated prompted feature $\mathbf{X}^{(\mathcal{P})} \in \mathbb{R}^{N \times F}$ as

$$\mathbf{x}_i^{\mathcal{P}} = \mathbf{x}_i + \sum_{s \in \{0,1\}} \alpha_{1,s} \mathbf{p}_s, \quad , i = 1, \ldots, N, \tag{7}$$

where the coefficients are defined as

$$\alpha_{i,s} = \frac{\exp \widetilde{\alpha}_{i,s}}{\exp \widetilde{\alpha}_{i,0} + \exp \widetilde{\alpha}_{i,1}}, \quad \widetilde{\alpha}_{i,s} = -\text{Tanh}\left(\text{Sim}\left(\mathbf{x}_j, \mathbf{p}_k\right)\right). \tag{8}$$

That is to say, node $v_i$ is more inclined to aggregate feature from different sensitive group while simultaneously disperse the feature of the same sensitive group.

To better understand our design, consider an example where the sensitive attribute of the central node $v_i$ is 0. According to the proposed generation mechanism, after message passing between $v_i$ and the fixed tokens, $v_i$ will exhibit sensitive characteristic of nodes with the different sensitive attribute. In other words, we obtain a new feature $\mathbf{x}'$ from $\mathbf{x}$, where $\mathbf{x}'_i$ carries sensitive information from the opposite sensitive group $\mathcal{S}_1$. This process is similar to counterfactual data augmentation (CAD), which involves generating new data by making changes to the sensitive attributes of existing data. However, our approach has a key difference from traditional CAD. In practice, it is unrealistic to perform CAD for all nodes, as we often only have access to the sensitive attribute for a limited number

of them. To overcome this, we take an alternative approach: instead of changing the node's sensitive attribute directly, we adjust its sensitive representation to align with the opposite sensitive group. This allows us to generate counterfactual data without needing to know each node's sensitive attribute.

*4.1.2 Learnable Graph Prompts.* Inspired by the recently proposed graph prompt literature [9], we introduce learnable graph prompts $Q = \{q_1, \ldots, q_T\}$ to bridge the gap between the pre-training task and the downstream task. Specifically, each prompt $q_k$ is assigned with a learnable vector $\mathbf{q}_k$ and the generated prompted features $\mathbf{X}^{(Q)} \in \mathbb{R}^{N \times F}$ as

$$\mathbf{x}_i^{(Q)} = \mathbf{x}_i + \sum_{t=1}^{T} \beta_{1,t} \mathbf{q}_t, \quad i = 1, \ldots, N \tag{9}$$

where the coefficients are calculated as

$$\beta_{i,t} = \frac{\exp \widetilde{\beta}_{i,t}}{\sum_{j=1}^{T} \exp \widetilde{\beta}_{i,j}}, \quad \widetilde{\beta}_{i,t} = \exp\left(\mathbf{x}_i^{\top} \eta_t\right). \tag{10}$$

Here $\eta_t \in \mathbb{R}^F$ is learnable. This design can be effectively adapted to graphs with varying scales (i.e., different node numbers), and it optimizes storage efficiency for large-scale input graphs, having $O(T)$ learnable parameters [9].

## 4.2 Heterophily-Enhanced Edge Modification

Recent research has shown that GNNs tend to perform worse on fairness homophilic graphs (with high $h_{\mathcal{G}}$) and better on heterophilic graphs (with low $h_{\mathcal{G}}$), both in terms of accuracy and fairness metrics [21]. The reason is that, in graphs with high fairness homophily, nodes primarily exchange information with others from the same sensitivity group. This can widen the feature gap between different sensitivity groups, ultimately increasing bias. Therefore, reducing fairness homophily may lead to more equitable representations and better accuracy.

However, a potential issue with the hybrid graph prompts is that we only modify the features without altering the sensitive attributes. This means that the feature generation does not influence whether connected node pairs belong to the same sensitive group, thus not changing fairness homophily. This differs from traditional counterfactual data augmentation, where the sensitive attribute of each node is known and can be directly flipped (e.g., from 0 to 1) to create counterfactual samples. In such cases, if two nodes belong to the same group, changing their sensitive attributes can make them belong to different groups, thus reducing the fairness homophily. However, this approach often requires knowledge of each node's sensitive attribute, which is impractical due to privacy concerns.

To address this challenge, we propose a heterophily-enhanced edge modification strategy that transforms the original graph structure into one with higher heterophily. Specifically, we compute the probability of a node $v_i$ belonging to either of the two sensitive groups $\mathcal{S}_0$ or $\mathcal{S}_1$ by calculating the distance between the node and each sensitive group. This allows us to infer the likely sensitive group of each node. We then modify the graph structure to obtain $\mathbf{A}^{(FPrompt)}$, increasing heterophily (i.e., reducing homophily) by modifying edges according to the identified sensitive groups. We provide theoretical analysis of heterophily-enhanced edge modification in Section 5.

*4.2.1 Sensitive Group Assignment.* Following [10], we define $\mathbf{X}_k = [\mathbf{x}_s] \in \mathbb{R}^{|\mathcal{S}_k^L| \times F}, s \in |\mathcal{S}_k^L|$ as the representation matrix for sensitive group $\mathcal{S}_k^L$, where $\mathcal{S}_k^L$ denotes the training set with sensitive attribute $k$. In this way, the distance between node $v_i$ and $\mathcal{S}_k$ can be calculated as

$$\text{DIS}(v_i, \mathcal{S}_k) = \mathbf{x}_i - \text{PROG}(v_i, \mathcal{S}_k) \quad (11)$$

where $\text{PROG}(v_i, \mathcal{S}_k)$ denotes the projection of $v_i$ to $\text{Span}(\mathbf{X}_k)$ and $\text{Span}(\mathbf{X}_k)$ denotes the space spanned by the row vectors of $\mathbf{X}_k$. We estimate the projection via ridge regression as

$$\text{PROG}(v_i, \mathcal{S}_k) \approx \text{PROG}(v_i, \mathcal{S}_k^L) = \mathbf{x}_i - \gamma \mathbf{X}_k^\top \left( \mathbf{I} + \gamma \mathbf{X}_k \mathbf{X}_k^\top \right)^{-1} \mathbf{X}_k \mathbf{x}_i. \quad (12)$$

The probability that node $v_i$ belongs to $\mathcal{S}_k$ can be obtained by feeding $\text{DIS}(v_i, \mathcal{S}_k)$ into a softmax function. We assign $v_i$ to the sensitive group with the higher probability, and denote its sensitive attribute as $\hat{s}_i$ accordingly.

*4.2.2 Fair Edge Mask.* With estimated sensitive attributes, we propose to modify the subgraph $\mathcal{G}$ via a mask matrix as $\mathbf{A}^{(FPrompt)} = \mathbf{A} \circ \mathbf{E}$ where recall that $\mathbf{A}$ is the adjacency matrix of $\mathcal{G}$. Specifically, each entry $\mathbf{E}_{ij}$ of the mask matrix is sampled from a binomial distribution $\mathcal{B}(\epsilon)$ when $\hat{s}_i \neq \hat{s}_j$, and from $\mathcal{B}(1 - \epsilon)$ when $\hat{s}_i = \hat{s}_j$. Here $\epsilon$ is the probability of success. If $\epsilon = 0$, the matrix $\mathbf{E}$ retains all connections between nodes from the same sensitive group while removing all connections between nodes from different sensitive groups. Conversely, if $\epsilon = 1$, the opposite occurs. In other words, $\epsilon$ controls the retention of homophilic connections. To enhance information exchange between nodes of different sensitive groups, we set $\epsilon > 0.5$.

### 4.3 Prompt-Based Fine-Tunining

Given a pre-trained GNN model $\Psi$ and graph $\mathcal{G} = (\mathbf{A}, \mathbf{X})$, we apply the proposed prompting function $\Phi^{(FPrompt)}$ to obtained the generated features and adjacency matrix. Specifically, the generated features $\mathbf{X}^{(FPrompt)}$ is calculated according to Section 4.1 as

$$\mathbf{x}_i^{(FPrompt)} = \mathbf{x}_i + \sum_{s \in \{0,1\}} \alpha_{1,s} \mathbf{p}_s + \sum_{t=1}^{T} \beta_{1,t} \mathbf{q}_t, \quad , i = 1, \ldots, N. \quad (13)$$

The generated adjacency matrix $\mathbf{A}^{(FPrompt)}$ is obtained according to Section 4.2.2. With the prompted graph $\left( \mathbf{X}^{(FPrompt)}, \mathbf{A}^{(FPrompt)} \right)$, we obtain the output embedding as

$$\mathbf{Z} = \Xi \circ \Psi \left( \mathbf{X}^{(FPrompt)}, \mathbf{A}^{(FPrompt)} \right), \quad (14)$$

where recall that $\Xi$ denotes the adapter. We choose a two layer multilayer perceptron (MLP) as the adapter, and the output dimension equals to the class number. The total loss function is defined as

$$\mathcal{L} = \sum_{v_i \in \mathcal{V}^L} \ell_{CE}(\widetilde{\mathbf{z}}_i, y_i) + \lambda_1 \ell_{CE}(\mathbf{z}_i, y_i) + \lambda_2 \|\widetilde{\mathbf{z}}_i - \mathbf{z}_i\|_2^2. \quad (15)$$

where $\ell_{CE}$ represents the cross-entropy loss for node classification and $\widetilde{\mathbf{z}}_i$ denotes the features without counterfactual data augmentation, i.e.,

$$\widetilde{\mathbf{Z}} = \Xi \circ \Psi \left( \mathbf{X}^{(Q)}, \mathbf{A}^{(FPrompt)} \right), \quad (16)$$

where $\mathbf{X}^{(Q)}$ is defined in Section 4.1.2. In the loss function (15), the first and the second terms restrict the outputs to be aligned with

the downstream task. The third term aims to minimize the discrepancy between the representation $\widetilde{\mathbf{z}}_i$ and its augmented version $\mathbf{z}_i$, ensuring that the fine-tuning process ignores differences caused by varying sensitive attributes, thus promoting fairness. We freeze the parameters in the pre-trained GNN and only fine-tune the adapter and the learnable graph prompts defined in Section 4.1.2.

## 5 Theoretical Analysis

In this section, we theoretically analyze the effectiveness of FPrompt. Specifically, we show the adaptation ability in Section 5.1 and fairness guarantees in Section 5.2.

### 5.1 Adaptation Ability

Note that the prompting function of FPrompt is $\Phi^{(FPrompt)}(\mathcal{G}) = (\mathbf{A}^{(FPrompt)}, \mathbf{X}^{(FPrompt)})$. Assume that there exists another prompting function $\theta$, generating a graph template $\mathcal{G}^* = (\mathbf{A}^*, \mathbf{X}^*) = \theta(\mathcal{G})$ for a given downstream task. The candidate space for $\mathbf{A}^*$ and $\mathbf{X}^*$ is denoted as $\mathbb{A}$ and $\mathbb{X}$, respectively. Then we have the following theorem.

THEOREM 1. *Given a pre-trained GNN model $\Psi$, for any prompted graph $\widehat{\mathcal{G}} = (\hat{\mathbf{A}} \in \mathbb{A}, \hat{\mathbf{X}} \in \mathbb{X})$ in the candidate space of the graph template $\mathcal{G}^* = \theta(\mathcal{G})$, there exists a learnable graph prompt $\widehat{Q}$ in Section 4.1.2 satisfying*

$$\Psi \left( \Phi^{(FPrompt)}(\mathcal{G}) \right) = \Psi \left( \widehat{\mathcal{G}} \right). \quad (17)$$

Theorem 1 is a direct extension of [Theorem 1, [9]] and we omit the proof here. Theorem 1 suggests that for any prompting function, FPrompt can attain the theoretical upper bound of performance. This also holds when some prompting function generates an optimal graph template according to the downstream loss, which means that FPrompt can also achieve the same performance theoretically. As a result, FPrompt provides universal adaptability for handling pre-trained GNN models across various pre-training strategies.

### 5.2 Fairness Guarantee

For binary node classification task, we use the sigmoid function in the adapter to generate the prediction $\mathbf{z} = \Xi \circ \Psi(\mathbf{X}, \mathbf{A}) \in \mathbb{R}^N$, where $z_i$ indicates the probability that $v_i$ is classified as 1. We then introduce a quantitative criterion of fairness:

$$\Delta_{\text{GSP}}(\mathbf{z}) = \|\mathbb{E}[z_i \mid i \in \mathcal{S}_0] - \mathbb{E}[z_i \mid i \in \mathcal{S}_1]\|. \quad (18)$$

It can be viewed as a generalization of the commonly used metric statistical parity (Definition 3.1) [18], considering that

$$\begin{aligned} \mathbb{E}[z_i \mid i \in \mathcal{S}_0] &= \int z_i \mathbb{P}(s_i = 0) dz_i \\ &= \int \mathbb{P}(\hat{y}_i = 1 \mid z_i) \mathbb{P}(z_i \mid s_i = 0) dz_i, \end{aligned} \quad (19)$$

where the last term is exactly $\mathbb{P}(\hat{y}_i \mid s_i = 0)$. Note that $\Delta_{\text{GSP}}$ measures the disparity between the predictions for different sensitive groups, and a smaller $\Delta_{\text{GSP}}$ indicates less bias in the predictions.

Our goal is to provide an upper bound of $\Delta_{\text{GSP}}$. Before that, we make the following assumption:

*Assumption 1.* The activation functions in both the GNN backbone and the adapter exhibit Lipschitz continuity.

**Table 1: Detailed statistics of the datasets.**

| Statistic | $|\mathcal{V}|$ | $|\mathcal{E}|$ | $F$ | Label | Sensitivity |
|---|---|---|---|---|---|
| Credit | 30,000 | 1,436,858 | 13 | Future default | Age |
| Pokec_z | 67,797 | 882,765 | 277 | Working field | Region |
| Pokec_n | 66,569 | 729,129 | 266 | Working field | Region |
| Pokec_TR | 11,294 | 44,884 | 241 | Working field | Region |
| Pokec_BA | 7,949 | 32,008 | 241 | Working field | Region |
| Pokec_KO | 13,280 | 55,568 | 241 | Working field | Region |
| Pokec_PR | 41,203 | 499,304 | 241 | Working field | Region |

*Assumption* 2. *The aggregation function in GNN backbone assigns equal weight to each node in the neighbor set.*

Assumption 1 is easily satisfied, for instance, by activation functions such as ReLU, LeakyReLU, and Tanh. Assumption 2 encompasses a wide range of GNN models, including GraphSAGE, GIN, and GCN. We consider the adapter to be a MLP, which is widely used in many pre-training and fine-tuning graph model. For simplicity, we assume the both GNN backbone and the adapter have one layer, and our theoretical analysis can be easily extended to multi-layer cases (see Appendix C).

With the conditions established, we are now ready to present the main theorem in the following:

THEOREM 2. *For a pre-training and fine-tuning model $\Xi \circ \Psi :$ $(\mathbf{X}, \mathbf{A}) \to \mathbf{z} \in \mathbb{R}^N$, if Assumptions 1 and 2 hold, then we have*

$$\Delta_{GSP}(\mathbf{z}) \leq \|\mathbf{W}_\Xi\| \, \|\mathbf{W}_\Psi\| \left( \left( \overline{\mu}_0 + \overline{\mu}_1 - 1 \right) \Delta_{GSP}(\mathbf{X}) + 2\sqrt{N}\delta \right), \quad (20)$$

*where $\mathbf{W}_\Xi$ and $\mathbf{W}_\Psi$ are the parameters of the adapter and GNN backbone, respectively. $\overline{\mu}_k$ is the average homophily ratio of sensitive group $\mathcal{S}_k$ as $\overline{\mu}_k = \sum_{i \in \mathcal{S}_k} \mu_i / |\mathcal{S}_k|$, and*

$$|\delta| \leq \max\left(\delta_0, \delta_1\right), \delta_k^{(l+1)} = \max_{i \in \mathcal{S}_{1-k}} \left\| \mathbf{x}_i^{(l+1)} - \overline{\mathbf{c}}_k^{(l+1)} \right\|, \quad (21)$$

*where $\overline{\mathbf{c}}_k^{(l+1)} = \sum_{i \in \mathcal{S}_k} \mathbf{x}_i^{(l)} / |\mathcal{S}_k|, \quad k = 0, 1.$*

PROOF. The proof is left to Appendix C. □

From Theorem 2, we observe that the upper bound of $\Delta_{GSP}(\mathbf{z})$ is determined by the following key parts.

**Average homophily ratio.** A smaller $\overline{\mu}_k$ plays a critical role in reducing the upper bound of $\Delta_{GSP}(\mathbf{z})$, thereby contributing significantly to the model fairness. In Section 4.2, we present an edge modification strategy specifically designed to address this by altering the structure of the graph. The core idea behind this strategy is to promote connections between nodes belonging to different sensitive groups while simultaneously reducing the number of edges between nodes within the same sensitive group. By increasing cross-group connections and limiting within-group interactions, the modification lowers the expected value of $h_i$ for individual nodes. Since $h_i$ directly impacts $\overline{\mu}_k$, decreasing $h_i$ leads to a reduction in $\overline{\mu}_k$. Consequently, this effectively restructures the graph such that mitigates the effects of fairness homophily.

**Representation discrepancy between two sensitive groups.** Both $\Delta_{GSP}(\mathbf{X})$ and $\delta$ are key measures related to the representation discrepancy between two sensitive groups. As highlighted in Section 4.1.1, the introduction of a fixed graph prompt in conjunction with the original graph can be likened to the generation of counterfactual data. As discussed in Section 4.1.1, the introduction of a fixed graph prompt in connection with the original graph is akin to generating counterfactual data, which reduces group discrepancy. Therefore, this approach helps align the representations of different sensitive groups, ultimately shrinking the distance between them and promoting fairness. By introducing this fixed graph prompt, the model essentially reshapes the data space, creating a more balanced representation of the two groups. This alignment process minimizes differences between their respective learned representations. As a result, the distance between the representations of different sensitive groups is significantly diminished, which directly contributes to improved fairness in the model. This approach ensures that both groups are treated more equally by the model, and the overall impact of sensitive attributes on the learned representations is minimized, fostering a more equitable predictive process.

## 6 Experiments
## 6.1 Experimental Settings

*6.1.1 Datasets.* We compare our methods with other approaches on three public datasets as follows: 1) **Credit** [1]: the nodes in the dataset are clients and two nodes are connected if they have a high similarity of the credit accounts. The task is to classify the credit risk level as high or low with the sensitive attribute gender; 2) **Pokec_z** and **Pokec_n** [4]: both datasets are sampled from an anonymized version of the Pokec network of 2012 (a social network from Slovakia), where nodes correspond to users who live in two major regions and the region information is utilized as the sensitive attribute. The working field of the users is binarized and utilized as the labels in node classification. We summarize these datasets in Table 1 and the training/validation/testing split in Appendix A.1.

Furthermore, existing prompt fine-tuning methods often utilize singular value decomposition (SVD) to align the feature dimensions when validating the performance of the same pre-trained model across different datasets [28]. However, applying SVD may distort the sensitive attributes to features that lack actual semantic meaning, potentially influencing fairness research. To better evaluate the cross-dataset performance and fairness of pre-trained models, we construct a new benchmark. The benchmark consists of four datasets, all created by sampling from the Pokec social network data based on geographic regions [30]. We select the regions of Trenciansky, Banskobystricky, Presovsky, and Kosicky, which we refer to as Pokec_TR, Pokec_BA, Pokec_PR, and Pokec_KO, respectively. For more details we refer to Appendix A.2. In all datasets, the sensitive attribute is region, and the label is working field.

*6.1.2 Baselines.* Compared approaches are from five categories: **1) Vanilla GNNs**: GCN [17] is a widely used spectral GNN; **2) Fairness-aware GNNs**: FairGNN [4] uses adversarial training to achieve fairness on graphs; NIFTY [1] flips the sensitive attributes to get counterfactual data; FairVGNN [33] introduces a feature

**Table 2: Performance comparison of graph representation learning methods with respect to prediction and fairness. The backbone is GCN and pre-training strategy is Infomax. The best results are bold and the second best results are underlined.**

| Method | Credit | | | | Pokec_z | | | | Pokec_n | | | |
|---|---|---|---|---|---|---|---|---|---|---|---|---|
| | ACC (↑) | AUC (↑) | DP (↓) | EO (↓) | ACC (↑) | AUC (↑) | DP (↓) | EO (↓) | ACC (↑) | AUC (↑) | DP (↓) | EO (↓) |
| GCN | $70.92_{0.44}$ | $67.58_{0.54}$ | $14.45_{4.13}$ | $14.14_{4.89}$ | $68.79_{1.11}$ | $69.09_{1.21}$ | $6.38_{1.63}$ | $5.30_{1.35}$ | $67.91_{0.71}$ | $68.61_{0.57}$ | $1.83_{1.15}$ | $2.28_{2.11}$ |
| FairGNN | $72.85_{0.12}$ | $67.78_{0.76}$ | $9.44_{1.29}$ | $7.98_{7.79}$ | $67.89_{0.27}$ | $69.18_{0.40}$ | $2.20_{1.50}$ | $1.42_{1.14}$ | $68.70_{0.25}$ | $68.36_{0.25}$ | $\underline{1.33_{0.59}}$ | $1.57_{0.76}$ |
| NIFTY | $71.94_{3.16}$ | $67.47_{0.77}$ | $10.32_{2.71}$ | $8.43_{2.58}$ | $66.72_{0.44}$ | $66.87_{0.48}$ | $5.48_{1.75}$ | $2.90_{0.59}$ | $68.32_{0.45}$ | $68.07_{0.44}$ | $1.60_{1.16}$ | $1.59_{1.17}$ |
| FairVGNN | $77.19_{0.45}$ | $67.61_{0.62}$ | $10.28_{1.39}$ | $6.87_{2.53}$ | $68.86_{0.17}$ | $74.17_{0.38}$ | $2.99_{1.50}$ | $2.77_{1.02}$ | $66.86_{0.76}$ | $68.76_{0.67}$ | $5.62_{1.83}$ | $4.59_{1.62}$ |
| Infomax | $70.77_{2.06}$ | $67.54_{2.97}$ | $12.44_{3.30}$ | $10.94_{3.00}$ | $68.05_{0.61}$ | $70.62_{1.16}$ | $6.44_{2.57}$ | $4.61_{2.67}$ | $68.22_{0.42}$ | $68.12_{0.60}$ | $6.32_{2.22}$ | $4.77_{2.43}$ |
| GPF | $74.11_{3.52}$ | $67.10_{1.32}$ | $13.92_{3.52}$ | $13.34_{2.64}$ | $\mathbf{70.50_{0.23}}$ | $76.66_{0.40}$ | $11.30_{0.96}$ | $10.92_{2.14}$ | $68.83_{0.91}$ | $67.62_{0.77}$ | $9.29_{1.34}$ | $6.20_{2.25}$ |
| GraphPrompt | $73.19_{1.34}$ | $68.37_{0.80}$ | $15.95_{3.09}$ | $14.13_{3.20}$ | $68.81_{0.24}$ | $65.55_{1.14}$ | $9.31_{0.11}$ | $8.69_{1.05}$ | $67.65_{0.64}$ | $67.54_{0.14}$ | $8.18_{1.25}$ | $7.78_{0.25}$ |
| GraphPAR | $74.83_{2.32}$ | $\underline{69.46_{0.43}}$ | $\underline{7.53_{2.62}}$ | $\underline{6.03_{3.13}}$ | $66.69_{1.41}$ | $73.40_{0.53}$ | $\underline{1.84_{0.65}}$ | $\underline{1.36_{0.56}}$ | $\underline{69.01_{0.88}}$ | $\mathbf{74.79_{1.01}}$ | $1.61_{0.31}$ | $\underline{1.52_{1.05}}$ |
| FPrompt | $\mathbf{77.42_{2.07}}$ | $\mathbf{70.26_{0.85}}$ | $\mathbf{4.34_{2.04}}$ | $\mathbf{3.05_{2.30}}$ | $\underline{68.94_{0.57}}$ | $\mathbf{76.70_{1.28}}$ | $\mathbf{1.56_{1.04}}$ | $\mathbf{0.86_{0.54}}$ | $\mathbf{69.54_{0.47}}$ | $\underline{74.63_{1.05}}$ | $\mathbf{0.78_{0.24}}$ | $\mathbf{1.28_{1.11}}$ |

masking strategy to prevent sensitive information leakage during the feature propagation; **3) Pre-training with fine-tuning**: Infomax [32] maximizes the mutual information between node and graph representations; GRACE [35] perturbs the graph model parameter spaces and narrow down the gap between different perturbations for the same graph; **4) Pre-training with prompt fine-tuning**: GPF [9] adds soft prompts to all node features of the input graph; GraphPrompt [20] inserts the prompt vector into the graph pooling by element-wise multiplication;; **5) Fairness-aware pre-training with fine-tuning**: GraphPAR [41] introduces a sensitive semantic augmenter that incorporates varying sensitive attribute semantics for each node. For all pre-training with and without prompt fine-tuning models, we choose the same adapter as FPrompt, which is a 2-layer MLP with output dimension equal to the class number.

*6.1.3 Implements.* We use the library PygDebias[1] to implement GCN, FairGNN, NIFTY, and FairVGNN. We apply the library ProG[2] for pre-training with (prompt) fine-tuning models including Infomax, GRACE, GPF, and GraphPrompt. For GraphPAR, we use the source code at Github[3]. For our model FPrompt, we search hyperparameters by the grid search method and we refer to Appendix A.3 for detailed explanation.

## 6.2 Prediction Performance and Fairness

We present the results across different datasets in Fig. 2. Our observations reveal that while vanilla GNNs, as well as those utilizing pre-training with fine-tuning and pre-training with prompt fine-tuning, achieve commendable accuracy performance, they exhibit significant shortcomings when it comes to fairness metrics. This indicates that although these models can classify or predict outcomes accurately, they often do so at the expense of equitable treatment across different demographic groups. In contrast, fairness-aware GNNs demonstrate strong fairness performance. However, a notable drawback of these fairness-focused models is their requirement for separate training of GNNs on each dataset. This separation can lead to inefficiencies in practical applications. FPrompt achieves

---

[1]https://github.com/yushundong/PyGDebias
[2]https://github.com/sheldonresearch/ProG
[3]https://github.com/BUPT-GAMMA/GraphPAR

**Table 3: Performance comparison with respect to prediction and fairness. The backbone is GCN and pre-training strategy is GRACE. The best results are bold and the second best results are underlined.**

| Method | Credit | | | |
|---|---|---|---|---|
| | ACC (↑) | AUC (↑) | DP (↓) | EO (↓) |
| GRACE | $71.57_{2.50}$ | $68.12_{3.58}$ | $9.03_{2.21}$ | $7.61_{2.49}$ |
| GPF | $74.25_{2.71}$ | $67.96_{3.32}$ | $11.72_{4.26}$ | $9.35_{3.24}$ |
| GraphPrompt | $74.70_{2.68}$ | $67.89_{2.98}$ | $9.91_{3.17}$ | $9.04_{2.58}$ |
| GraphPAR | $\underline{75.52_{2.15}}$ | $68.23_{2.44}$ | $\mathbf{5.62_{1.64}}$ | $\mathbf{4.25_{1.58}}$ |
| FPrompt | $\mathbf{76.35_{2.68}}$ | $\mathbf{69.35_{3.68}}$ | $\underline{5.01_{2.68}}$ | $\underline{4.55_{1.56}}$ |

**Table 4: Performance comparison with respect to prediction and fairness. The pre-training strategy is Infomax. The best results are bold and the second best results are underlined.**

| Method | | Credit | | | |
|---|---|---|---|---|---|
| | | ACC (↑) | AUC (↑) | DP (↓) | EO (↓) |
| GAT | Infomax | $70.55_{3.81}$ | $67.45_{1.24}$ | $12.06_{4.21}$ | $11.42_{4.35}$ |
| | GPF | $\underline{75.26_{2.95}}$ | $68.89_{1.85}$ | $12.81_{4.53}$ | $12.20_{3.56}$ |
| | GraphPAR | $75.21_{2.18}$ | $\mathbf{69.54_{0.24}}$ | $\underline{7.75_{1.86}}$ | $\underline{6.35_{2.39}}$ |
| | FPrompt | $\mathbf{76.18_{1.74}}$ | $\underline{69.53_{1.49}}$ | $\mathbf{5.94_{3.02}}$ | $\mathbf{4.77_{2.50}}$ |
| GIN | Infomax | $70.29_{3.66}$ | $65.92_{3.20}$ | $10.66_{2.27}$ | $10.88_{3.40}$ |
| | GPF | $74.71_{2.99}$ | $\underline{69.81_{3.24}}$ | $11.57_{3.62}$ | $8.81_{2.71}$ |
| | GraphPAR | $\underline{74.94_{5.19}}$ | $68.86_{2.87}$ | $\underline{6.07_{2.33}}$ | $\underline{4.42_{2.12}}$ |
| | FPrompt | $\mathbf{76.06_{1.15}}$ | $\mathbf{69.71_{3.35}}$ | $\mathbf{4.90_{1.80}}$ | $\mathbf{3.78_{1.31}}$ |

both strong accuracy and fairness performance simultaneously. For example, FPrompt significantly enhances fairness performance on the Credit, Pokec_z, and Pokec_n datasets, with improvements of 42%, 15%, and 41%, respectively, in terms of demographic parity.

**Table 5: Performance comparison of graph representation learning methods with respect to prediction and fairness. The backbone is GCN and pre-training strategy is Infomax. The best results are bold and the second best results are underlined.**

| Method | Pokec_TR → Pokec_BA | | | | Pokec_TR → Pokec_KO | | | | Pokec_TR → Pokec_PR | | | |
|---|---|---|---|---|---|---|---|---|---|---|---|---|
| | ACC (↑) | AUC (↑) | DP (↓) | EO (↓) | ACC (↑) | AUC (↑) | DP (↓) | EO (↓) | ACC (↑) | AUC (↑) | DP (↓) | EO (↓) |
| Infomax | $71.99_{0.63}$ | $75.11_{0.22}$ | $7.60_{3.63}$ | $5.29_{1.81}$ | $65.27_{1.49}$ | $66.70_{2.36}$ | $3.46_{2.47}$ | $5.58_{2.22}$ | $66.34_{0.24}$ | $\underline{68.94_{0.58}}$ | $24.97_{2.08}$ | $25.89_{2.38}$ |
| GPF | $\underline{72.63_{0.71}}$ | $76.70_{0.44}$ | $9.09_{3.62}$ | $10.46_{2.75}$ | $\mathbf{67.36_{1.52}}$ | $\underline{68.34_{1.32}}$ | $4.56_{1.62}$ | $6.64_{1.83}$ | $66.17_{0.18}$ | $67.93_{0.92}$ | $28.73_{4.87}$ | $27.56_{5.28}$ |
| GraphPrompt | $72.12_{1.67}$ | $\mathbf{76.98_{1.56}}$ | $8.82_{4.58}$ | $8.56_{2.91}$ | $65.99_{1.29}$ | $66.14_{2.58}$ | $4.98_{1.18}$ | $6.54_{2.15}$ | $65.62_{0.51}$ | $67.79_{1.18}$ | $25.32_{2.53}$ | $25.78_{2.57}$ |
| GraphPAR | $71.00_{2.54}$ | $76.45_{1.21}$ | $\underline{4.97_{2.85}}$ | $\underline{4.65_{1.87}}$ | $65.30_{1.71}$ | $66.26_{1.54}$ | $\underline{3.45_{2.12}}$ | $\underline{2.55_{2.06}}$ | $\underline{66.54_{0.51}}$ | $\mathbf{69.06_{0.79}}$ | $\underline{15.70_{3.94}}$ | $\underline{14.10_{3.86}}$ |
| FPrompt | $\mathbf{72.81_{0.86}}$ | $\underline{76.91_{1.68}}$ | $\mathbf{4.57_{2.90}}$ | $\mathbf{3.24_{2.46}}$ | $\underline{66.48_{2.81}}$ | $\mathbf{69.03_{1.87}}$ | $\mathbf{1.70_{0.62}}$ | $\mathbf{2.10_{1.50}}$ | $\mathbf{66.62_{5.58}}$ | $67.95_{0.55}$ | $\mathbf{8.93_{3.67}}$ | $\mathbf{6.52_{3.36}}$ |

To further demonstrate FPrompt's ability to enhance different pre-trained strategies, we experiment with the pre-training method GRACE.The results, as outlined in Table 3, clearly illustrate that FPrompt effectively reduces bias in the pre-trained model, leading to more balanced outcomes across different demographic groups. Moreover, we extend our evaluation to include different GNN architectures, as presented in Table 4. The results indicate that FPrompt adapts well to various GNN architectures, including GIN (which satisfies Assumption 2) and GAT (which does not). This adaptability underscores the versatility of FPrompt. For more experimental results, please refer to Appendix B.

### 6.3 Transferability Analysis

To assess the transferability of our model, we compare it against both the fine-tuning method Infomax and various prompt-based fine-tuning methods. We intentionally exclude fairness-aware GNNs from this comparison, as these models require retraining the GNN on each new dataset, which limits their adaptability in transfer learning scenarios. Our evaluation is conducted using our newly proposed benchmark, where we pre-train the GNN model on the Pokec_TR dataset and subsequently fine-tune it on different downstream datasets, applying different fine-tuning strategies. The results of this comparison are summarized in Table 5. We observe that while prompt-based fine-tuning methods tend to achieve high accuracy, they generally perform poorly in terms of fairness metrics. In contrast, FPrompt not only maintains competitive accuracy but also significantly improves fairness outcomes across datasets, demonstrating its ability to in transfer learning tasks.

### 6.4 Model Analysis

To further verify the theoretical effectiveness of our method as discussed in Section 5.2, we present a detailed comparison of the representation discrepancies between two sensitive groups (i.e., $\Delta_{\text{GSP}}(\mathbf{X})$ and $\delta$). These discrepancies are evaluated both for the raw features and after applying our proposed FPrompt method. In particular, we observe in Figure 2 that the use of a fixed graph prompt within FPrompt significantly reduces both $\Delta_{\text{GSP}}(\mathbf{X})$ and $\delta$. This reduction in discrepancies between sensitive groups indicates that the representations of individuals from different sensitive categories become more aligned, thereby reducing bias. This result aligns closely with our theoretical analysis, which suggests that FPrompt's ability to reduce the discrepancy between sensitive groups can directly contribute to enhancing fairness in graph-based

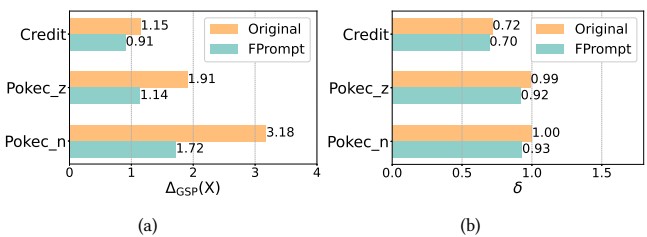

(a)           (b)

**Figure 2: The representation discrepancy between the two sensitive groups. (a): The generalized SP; (b): The maximum feature distance of the raw features (denoted as Original) and prompted features $\mathbf{X}^{\mathcal{P}}$ (denoted as FPrompt). We unify the scale for better presentation.**

predictions. By narrowing the gap between group representations of different sensitive groups, FPrompt ensures that individuals are treated more equitably in downstream tasks.

### 7 Conclusion

In this work, we proposed FPrompt, a fairness-aware graph prompt tuning method designed to mitigate the inherent bias present in pre-trained graph neural networks (GNNs) while retaining their generalization capabilities. By introducing hybrid graph prompts for counterfactual data augmentation and applying edge modification to increase sensitivity heterophily, FPrompt successfully aligns pre-training and downstream tasks, addressing bias introduced by sensitive attributes. Theoretical analysis demonstrate two key aspects of FPrompt: (1) it has universal capabilities to effectively adapt to various pre-training strategies, and (2) it reduces the upper bound of generalized statistical parity, significantly mitigating bias in pre-trained models. Extensive experiments confirm that FPrompt outperforms existing methods in both fairness and accuracy on standard benchmarks.

The present work also opens up promising future directions. One notable opportunity arises from the success of prompt tuning in adapting pre-trained GNN models to tasks beyond node classification (e.g., link prediction). However, bias can arise in link prediction due to factors such as imbalanced data, where certain types of links (e.g., between nodes from different sensitive groups) are underrepresented. Therefore, a potential next step would be to explore the application of FPrompt to reduce bias of pre-trained models in link prediction tasks.

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

## A Experimental Details

### A.1 Datasets Split

In our experiments, we adhere to standard dataset splits and employ random seeds for re-producibility. Note that for all datasets, the node labels are binary (i.e., 0 and 1). Denote the node set with label 0 and 1 as $\mathcal{V}_0$ and $\mathcal{V}_1$, respectively. Then we randomly split the nodes into validation/testing sets with ratio $0.25|\mathcal{V}_0| + 0.25|\mathcal{V}_1|/0.25|\mathcal{V}_0| + 0.25|\mathcal{V}_1|$. The training ratio is set to

$$\min(T, 0.25|\mathcal{V}_0|) + \min(T, 0.25|\mathcal{V}_1|), \tag{22}$$

where we choose $T = 3000$ for all datasets. We also summarize other statistical details in Table 6.

### A.2 Benchmark Construction

We introduce the construction details of Pokec_PR as an example, and the remaining datasets follow the same process. We first filter out nodes where the region starts with "presovsky kraj" and select the following columns: *user_id, public, completion_percentage, gender, region, AGE, I_am_working_in_field, spoken_languages, hobbies, I_most_enjoy_good_food, body_type, eye_color, hair_color, hair_type, completed_level_of_education, favourite_color, relation_to_smoking, relation_to_alcohol, on_pokec_i_am_looking_for, love_is_for_me, relation_to_casual_sex, my_partner_should_be, marital_status, relation_to_children, I_like_movies, I_like_watching_movie, I_like_music, I_mostly_like_listening_to_music, the_idea_of_good_evening, I_like_specialties_from_kitchen, I_am_going_to_concerts, my_active_sports, my_passive_sports, and I_like_books.* For the features from *spoken_languages* onwards, we generate columns where the column names correspond to the values of those features. If an element has that value for the feature, the corresponding element is 1; otherwise, it is 0. Note that a feature can have multiple values for a single element. Then, following the existing data, we map I_am_working_in_field to 0, 1, and -1 (where -1 indicates missing data). Next, we select the two most frequent regions and filter the corresponding data as the final dataset (*presovsky kraj, poprad and presovsky kraj, bardejov*). Finally, we extract the connection relationships based on user_id. The source data can be found from [30].

### A.3 Implements

We conduct all the experiments on a machine with an NVIDIA A100 80GB PCIe, Intel Xeon CPU (2.20 GHz) with 6 cores, and 150 GB of RAM. Our code is built on pytorch==2.1.1, pyg==2.4.0, cuda==12.1, and dgl==2.1.0. We choose learning_rate=0.001, weight_decay=1e-5, and num_layer=2 for all datasets. For Credit, we set hidden_dim=16 and for the remaining datasets, we let hidden_dim=24. The optimal hyperparameters such as regularization term $\lambda_1, \lambda_2$, edge modifictaion ratio $\epsilon$, and learnable token number $T$ are tuned for each dataset separately. We will release the source code after the review process.

## B Additional Experimental Results

### B.1 Results of GNN Backbones

We conduct experiments on Pokec_z with different GNN backbones in Table 7. It can be observed that FPrompt outperforms baseline models in terms of classification performance and fairness.

### B.2 Results on Pokec_TR

We provide the results where we first pre-train a GNN model on Pokec_TR and fine-tune the model on the same dataset. Results in Table. 8 demonstrate the state-of-the-art performance of FPrompt.

## C Proof of Theorem 2

To prove Theorem 2, we first derive the upper bound of $\Delta_{\text{GSP}}$ for both the GNN backbone (Theorem 3) and the adapter (Theorem 4) separately. Afterward, the overall upper bound for the pre-training and fine-tuning graph model can be directly derived by combining these results.

### C.1 $\Delta_{\text{GSP}}$ for the GNN Backbone

For a GNN model that outputs a feature matrix rather than a prediction vector as Eq. (18), the fairness criterion is defined as

$$\Delta_{\text{GSP}}(\mathbf{H}) = \|\mathbb{E}[\mathbf{h}_i \mid i \in \mathcal{S}_0] - \mathbb{E}[\mathbf{h}_i \mid i \in \mathcal{S}_1]\|, \tag{23}$$

where $\mathbf{h}_i$ is the $i$-th row of $\mathbf{H}$. We have the following result.

THEOREM 3. *For a GNN model* $\Psi : (\mathbf{X}, \mathbf{A}) \to \mathbf{H}$*, if Assumptions 1 and 2 hold, then we have*

$$\Delta_{GSP}\left(\mathbf{H}^{(l+1)}\right) \leq \left((\overline{\mu}_0 + \overline{\mu}_1 - 1)\Delta_{GSP}\left(\mathbf{H}^{(l)}\right) + 2\sqrt{N}\delta^{(l+1)}\right)\left\|\mathbf{W}_\Psi^{(l)}\right\|, \tag{24}$$

*where* $\mathbf{W}_\Psi^{(l)}$ *is the parameter at the l-th layer of GNN,* $\overline{\mu}_k$ *is the average homophily ratio of sensitive group* $\mathcal{S}_k$ *as* $\overline{\mu}_k = \sum_{i \in \mathcal{S}_k} \mu_i/|\mathcal{S}_k|$*, and*

$$|\delta^{(l+1)}| \leq \max\left(\delta_0^{(l+1)}, \delta_1^{(l+1)}\right), \delta_k^{(l+1)} = \max_{i \in \mathcal{S}_{1-k}}\left\|\mathbf{c}_i^{(l+1)} - \overline{\mathbf{c}}_k^{(l+1)}\right\|, \tag{25}$$

*where* $\overline{\mathbf{c}}_k^{(l+1)} = \sum_{i \in \mathcal{S}_k} \mathbf{h}_i^{(l)}/|\mathcal{S}_k|, \quad k = 0, 1.$

PROOF OF THEOREM 3. Estimating the expected distribution with empirical distribution, we have

$$\Delta_{\text{GSP}}(\mathbf{H}^{(l+1)}) = \left\|\mathbb{E}\left[\mathbf{h}_i^{(l+1)} \mid i \in \mathcal{S}_0\right] - \mathbb{E}\left[\mathbf{h}_i^{(l+1)} \mid i \in \mathcal{S}_1\right]\right\|$$

$$= \left\|\frac{1}{|\mathcal{S}_0|}\sum_{i \in \mathcal{S}_0}\mathbf{h}_i^{(l+1)} - \frac{1}{|\mathcal{S}_1|}\sum_{i \in \mathcal{S}_1}\mathbf{h}_i^{(l+1)}\right\|$$

$$= \left\|\frac{1}{|\mathcal{S}_0|}\sum_{i \in \mathcal{S}_0}\sigma\left(\mathbf{r}_i^{(l+1)}\right) - \frac{1}{|\mathcal{S}_1|}\sum_{i \in \mathcal{S}_1}\sigma\left(\mathbf{r}_i^{(l+1)}\right)\right\| \tag{26}$$

$$\overset{(a)}{\leq} \left\|\frac{1}{|\mathcal{S}_0|}\sum_{i \in \mathcal{S}_0}\mathbf{r}_i^{(l+1)} - \frac{1}{|\mathcal{S}_1|}\sum_{i \in \mathcal{S}_1}\mathbf{r}_i^{(l+1)}\right\|,$$

where $(a)$ is due to the Lipschitz continuity of the activation functions (Assumption 1). Here, $\sigma(\cdot)$ denotes the activation function and $\mathbf{r}_i^{(l+1)}$ is the representation of node $v_i$ after aggregation. The first term can be rewritten as

$$\frac{1}{|\mathcal{S}_0|}\sum_{i \in \mathcal{S}_0}\mathbf{r}_i^{(l+1)}$$

$$= \frac{1}{|\mathcal{S}_0|}\sum_{i \in \mathcal{S}_0}\frac{1}{|\mathcal{N}_i|}\left(\sum_{j \in \mathcal{N}_i \cap \mathcal{S}_0}\mathbf{c}_j^{(l+1)} + \sum_{j \in \mathcal{N}_i \cap \mathcal{S}_1}\mathbf{c}_j^{(l+1)}\right)$$

$$= \frac{1}{|\mathcal{S}_0|}\sum_{i \in \mathcal{S}_0}\frac{1}{|\mathcal{N}_i|}\left(\sum_{j \in \mathcal{N}_i \cap \mathcal{S}_0}\overline{\mathbf{c}}_1^{(l+1)} + \sum_{j \in \mathcal{N}_i \cap \mathcal{S}_1}\overline{\mathbf{c}}_0^{(l+1)}\right) + \delta^{(l+1)} \tag{27}$$

$$= \frac{1}{|\mathcal{S}_0|}\sum_{i \in \mathcal{S}_0}\left(\mu_i\overline{\mathbf{c}}_1^{(l+1)} + (1 - \mu_i)\overline{\mathbf{c}}_0^{(l+1)}\right) + \delta^{(l+1)}$$

$$= \overline{\mu}_0\overline{\mathbf{c}}_1^{(l+1)} + (1 - \overline{\mu}_0)\overline{\mathbf{c}}_0^{(l+1)} + \delta'^{(l+1)},$$

**Table 6: Experimental detail of the datasets.**

| Statistic | Training Size | Label Ratio | Label Distribution ($|\mathcal{V}_0|/|\mathcal{V}_1|$) | Sensitivity Distribution ($|\mathcal{S}_0|/|\mathcal{S}_1|$) |
|---|---|---|---|---|
| Credit | 6,000 | 1.00 | 0.28 | 10.17 |
| Pokec_z | 5,131 | 0.15 | 0.86 | 1.84 |
| Pokec_n | 4,398 | 0.13 | 1.05 | 2.46 |
| Pokec_TR | 3,746 | 0.66 | 0.97 | 1.08 |
| Pokec_BA | 2,314 | 0.58 | 1.41 | 1.00 |
| Pokec_KO | 4,002 | 0.60 | 1.67 | 1.41 |
| Pokec_PR | 3,780 | 0.18 | 1.15 | 1.68 |

**Table 7: Performance comparison of graph representation learning methods with respect to prediction and fairness. The pre-training strategy is Infomax. The best results are bold and the second best results are underlined.**

| Method | | ACC (↑) | AUC (↑) | DP (↓) | EO (↓) |
|---|---|---|---|---|---|
| | | | | Pokec_z | |
| GAT | Infomax | $66.39_{0.52}$ | $70.87_{1.20}$ | $6.51_{2.22}$ | $5.27_{2.30}$ |
| | GPF | $\underline{67.43_{0.36}}$ | $72.10_{0.74}$ | $10.36_{1.52}$ | $8.28_{2.66}$ |
| | GraphPAR | $67.20_{0.69}$ | $\underline{73.38_{0.95}}$ | $\underline{1.55_{0.90}}$ | $\underline{1.50_{1.14}}$ |
| | FPrompt | $\mathbf{67.97_{0.73}}$ | $\mathbf{73.91_{0.54}}$ | $\mathbf{1.42_{1.17}}$ | $\mathbf{1.29_{0.74}}$ |
| GIN | Infomax | $64.90_{0.65}$ | $71.23_{0.94}$ | $4.22_{3.15}$ | $4.06_{2.23}$ |
| | GPF | $\underline{66.72_{1.28}}$ | $\underline{73.25_{0.98}}$ | $10.25_{2.34}$ | $9.28_{3.51}$ |
| | GraphPAR | $65.49_{1.77}$ | $73.03_{1.32}$ | $\underline{3.74_{2.06}}$ | $\underline{3.04_{1.52}}$ |
| | FPrompt | $\mathbf{68.43_{0.72}}$ | $\mathbf{74.59_{0.72}}$ | $\mathbf{2.52_{1.14}}$ | $\mathbf{1.13_{0.92}}$ |

**Table 8: Performance comparison of graph representation learning methods with respect to prediction and fairness. The backbone is GCN and pre-training strategy is Infomax. The best results are bold and the second best results are underlined.**

| Method | ACC (↑) | AUC (↑) | DP (↓) | EO (↓) |
|---|---|---|---|---|
| | | Pokec_TR | | |
| Infomax | $\mathbf{76.23_{0.76}}$ | $\mathbf{80.96_{0.44}}$ | $3.68_{2.61}$ | $4.70_{2.20}$ |
| GraphPAR | $74.26_{0.56}$ | $79.13_{0.42}$ | $\underline{2.89_{1.39}}$ | $\underline{2.64_{1.38}}$ |
| FPrompt | $\underline{75.82_{2.29}}$ | $\underline{79.88_{0.45}}$ | $\mathbf{1.45_{1.40}}$ | $\mathbf{2.00_{0.92}}$ |

where $\mathbf{c}_i^{(l+1)} = \mathbf{W}_\Psi^{(l)}\mathbf{h}_i^{(l)}$, $\bar{\mu}_0$ is the average homophily ratio of sensitive group $\mathcal{S}_0$. $\bar{\mathbf{c}}_1^{(l+1)}$ denotes the average representation of sensitive group $\mathcal{S}_k$ at the $l$-th layer as

$$\bar{\mathbf{c}}_k^{(l+1)} = \frac{1}{|\mathcal{S}_k|} \sum_{i \in \mathcal{S}_k} \mathbf{h}_i^{(l)}, \quad k = 0, 1. \tag{28}$$

The error term $\delta'^{(l+1)}$ is upper-bounded by

$$|\delta'^{(l+1)}| \le \max\left(\delta_0^{(l+1)}, \delta_1^{(l+1)}\right), \tag{29}$$

with $\delta_k^{(l+1)} = \max_{i \in \mathcal{S}_{1-k}} \left\|\mathbf{c}_i^{(l+1)} - \bar{\mathbf{c}}_k^{(l+1)}\right\|$ denoting the maximum feature distance between node $v_i$ and the average representation of the opposite sensitive class. Similarly, the second term can be rewritten as

$$
\begin{aligned}
&\frac{1}{|\mathcal{S}_1|} \sum_{i \in \mathcal{S}_1} \mathbf{r}_i^{(l+1)} \\
&= \frac{1}{|\mathcal{S}_1|} \sum_{i \in \mathcal{S}_1} \frac{1}{|\mathcal{N}_i|} \left( \sum_{j \in \mathcal{N}_i \cap \mathcal{S}_0} \mathbf{c}_j^{(l+1)} + \sum_{j \in \mathcal{N}_i \cap \mathcal{S}_1} \mathbf{c}_j^{(l+1)} \right) \\
&= (1 - \bar{\mu}_1)\bar{\mathbf{c}}_1^{(l+1)} + \bar{\mu}_1 \bar{\mathbf{c}}_0^{(l+1)} + \delta''^{(l+1)},
\end{aligned}
\tag{30}
$$

where $\delta''^{(l+1)}$ has the same upper bound with $\delta'^{(l+1)}$. As a result, we have

$$
\begin{aligned}
&\left\| \frac{1}{|\mathcal{S}_0|} \sum_{i \in \mathcal{S}_0} \mathbf{r}_i^{(l+1)} - \frac{1}{|\mathcal{S}_1|} \sum_{i \in \mathcal{S}_1} \mathbf{r}_i^{(l+1)} \right\| \\
&= (\bar{\mu}_0 + \bar{\mu}_1 - 1)\left\|\bar{\mathbf{c}}_0^{(l+1)} - \bar{\mathbf{c}}_1^{(l+1)}\right\| + 2\sqrt{N}\delta'''^{(l+1)},
\end{aligned}
\tag{31}
$$

where $\delta'''^{(l+1)}$ is also upper-bounded as in Eq. (29). Notice that

$$
\begin{aligned}
&\left\|\bar{\mathbf{c}}_0^{(l+1)} - \bar{\mathbf{c}}_1^{(l+1)}\right\| \\
&= \left\|\mathbf{W}_\Psi^{(l)}\right\| \left\| \frac{1}{|\mathcal{S}_0|} \sum_{i \in \mathcal{S}_0} \mathbf{h}_i^{(l)} - \frac{1}{|\mathcal{S}_1|} \sum_{i \in \mathcal{S}_1} \mathbf{h}_i^{(l)} \right\| \\
&= \left\|\mathbf{W}_\Psi^{(l)}\right\| \Delta_{\text{GSP}}\left(\mathbf{H}^{(l)}\right).
\end{aligned}
\tag{32}
$$

Combining Eq. (26), Eq. (27), and Eq. (32), we arrive that

$$\Delta_{\text{GSP}}\left(\mathbf{H}^{(l+1)}\right) \le \left( (\bar{\mu}_0 + \bar{\mu}_1 - 1)\Delta_{\text{GSP}}\left(\mathbf{H}^{(l)}\right) + 2\sqrt{N}\delta^{(l+1)} \right)\left\|\mathbf{W}_\Psi^{(l)}\right\|, \tag{33}$$

where

$$|\delta'^{(l+1)}| \le \max\left(\delta_0^{(l+1)}, \delta_1^{(l+1)}\right), \delta_k^{(l+1)} = \max_{i \in \mathcal{S}_{1-k}} \left\|\mathbf{c}_i^{(l+1)} - \bar{\mathbf{c}}_k^{(l+1)}\right\|. \tag{34}$$

Proof finished. □

## C.2 $\Delta_{\text{GSP}}$ for the Adapter

THEOREM 4. *For a L-layer MLP adapter $\Xi : \mathbf{H} \to \mathbf{z} \in \mathbb{R}^N$, if Assumption 1 holds, we have*

$$\Delta_{GSP}(\mathbf{z}) \leq \prod_{l=0,\dots,L-1} \left\| \mathbf{W}_{\Xi}^{(l)} \right\| \Delta_{GSP}(\mathbf{H}), \tag{35}$$

*where $\mathbf{W}_{\Xi}^{(l)}$ is the parameter at the $l$-th layer of the adapter.*

PROOF OF THEOREM 4. Denote the representation at the $l$-th layer as $\mathbf{Z}^{(l)}$ and $\mathbf{Z}^{(0)} = \mathbf{H}$. Notice that

$$\begin{aligned}
\Delta_{\text{GSP}}\left(\mathbf{Z}^{(l+1)}\right) &= \left\| \mathbb{E}\left[ \mathbf{z}_i^{(l+1)} \mid i \in \mathcal{S}_0 \right] - \mathbb{E}\left[ \mathbf{z}_i^{(l+1)} \mid i \in \mathcal{S}_1 \right] \right\| \\
&= \left\| \frac{1}{|\mathcal{S}_0|} \sum_{i \in \mathcal{S}_0} \mathbf{z}_i^{(l+1)} - \frac{1}{|\mathcal{S}_1|} \sum_{i \in \mathcal{S}_1} \mathbf{z}_i^{(l+1)} \right\| \\
&= \left\| \frac{1}{|\mathcal{S}_0|} \sum_{i \in \mathcal{S}_0} \sigma\left( \mathbf{t}_i^{(l+1)} \right) - \frac{1}{|\mathcal{S}_1|} \sum_{i \in \mathcal{S}_1} \sigma\left( \mathbf{t}_i^{(l+1)} \right) \right\| \\
&\overset{(a)}{\leq} \left\| \frac{1}{|\mathcal{S}_0|} \sum_{i \in \mathcal{S}_0} \mathbf{t}_i^{(l+1)} - \frac{1}{|\mathcal{S}_1|} \sum_{i \in \mathcal{S}_1} \mathbf{t}_i^{(l+1)} \right\|,
\end{aligned} \tag{36}$$

where $\mathbf{t}_i^{(l+1)} = \mathbf{W}_{\Xi}^{(l)} \mathbf{z}_i^{(l)}$, $\sigma(\cdot)$ denotes the activation function, and $(a)$ is due to the Lipschitz continuity of the activation functions

(Assumption 1). Because

$$\begin{aligned}
&\left\| \mathbf{t}_0^{(l+1)} - \mathbf{t}_1^{(l+1)} \right\| \\
&= \left\| \mathbf{W}_{\Xi}^{(l)} \right\| \left\| \frac{1}{|\mathcal{S}_0|} \sum_{i \in \mathcal{S}_0} \mathbf{z}_i^{(l)} - \frac{1}{|\mathcal{S}_1|} \sum_{i \in \mathcal{S}_1} \mathbf{z}_i^{(l)} \right\| \\
&= \left\| \mathbf{W}_{\Xi}^{(l)} \right\| \Delta_{\text{GSP}}\left( \mathbf{Z}^{(l)} \right).
\end{aligned} \tag{37}$$

Thus we have

$$\Delta_{\text{GSP}}(\mathbf{H}^{(l+1)}) \leq \left\| \mathbf{W}_{\Xi}^{(l)} \right\| \Delta_{\text{GSP}}(\mathbf{H}^{(l)}). \tag{38}$$

Proof finished. □

## C.3 $\Delta_{\text{GSP}}$ for the Pre-Training and Fine-Tuning Graph Model

PROOF OF THEOREM 2. For simplicity, we assume the both GNN backbone and the adapter have one layer. Thus we have

$$\begin{aligned}
\Delta_{\text{GSP}}(\mathbf{z}) &\leq \| \mathbf{W}_{\Xi} \| \Delta_{\text{GSP}}(\mathbf{H}) \\
&\leq \| \mathbf{W}_{\Xi} \| \| \mathbf{W}_{\Psi} \| \left( (\overline{\mu}_0 + \overline{\mu}_1 - 1)\Delta_{\text{GSP}}(\mathbf{X}) + 2\sqrt{N}\delta \right),
\end{aligned} \tag{39}$$

where

$$|\delta| \leq \max(\delta_0, \delta_1), \delta_k^{(l+1)} = \max_{i \in \mathcal{S}_{1-k}} \left\| \mathbf{x}_i^{(l+1)} - \overline{\mathbf{c}}_k^{(l+1)} \right\|, \tag{40}$$

where $\overline{\mathbf{c}}_k^{(l+1)} = \sum_{i \in \mathcal{S}_k} \mathbf{x}_i^{(l)} / |\mathcal{S}_k|, \quad k = 0, 1$. Proof finished. □

