# OpenReview forum: "Fairness-aware Prompt Tuning for Graph Neural Networks"
_ACM.org/TheWebConf/2025/Conference — WWW 2025 Poster_

### Official Review · Reviewer_wzRj · 2024-11-02

**Novelty:** 3
**Technical Quality:** 4

**Review:**

The paper proposes a fairness-aware prompt tuning approach called FPrompt. FPrompt addresses inherent biases in pre-trained GNNs by introducing hybrid graph prompts, which incorporate counterfactual data to promote fairness across sensitive groups, and edge modifications to increase message-passing between these groups. This method effectively reduces statistical bias in predictions while enhancing generalization performance across datasets. Experiments show that FPrompt outperforms other models in both fairness and accuracy, making it suitable for fairness-sensitive applications.

**Questions:**

Q1. How does the model ensure accuracy in identifying sensitive groups and modifying edges, and could further experiments clarify their independent contributions to fairness?

Q2. Node classification is the primary evaluation, but does FPrompt generalize to tasks like link prediction or subgraph classification that may require different fairness considerations?

Q3. To better understand each module's effectiveness, could ablation experiments and an analysis of the model’s scalability and computational efficiency be provided?

**Reviewer Confidence:**

3: The reviewer is confident but not certain that the evaluation is correct

**Scope:**

3: The work is somewhat relevant to the Web and to the track, and is of narrow interest to a sub-community

---

### Official Review · Reviewer_cM7x · 2024-11-22

**Novelty:** 4
**Technical Quality:** 3

**Review:**

# Strength
* The framework innovatively integrates **fairness constraints with graph prompt tuning**, targeting biases inherent in pre-trained GNNs.
* Extensive experiments on real-world datasets demonstrate that FPrompt outperforms state-of-the-art methods in terms of fairness and predictive performance.
* A new benchmark evaluates cross-dataset generalization, showing that FPrompt effectively balances accuracy and fairness in transfer learning tasks.

# Weakness

* While the topic is compelling, the novelty of the Hybrid Graph Prompts is limited, as it closely resembles GPF [1]. The authors should discuss the similarities and differences more explicitly.
* The theoretical contribution of Theorem 1 is derived from [1], yet the statement in the abstract (not directly) implies that it is a novel contribution. The authors should revise this to accurately reflect the source of the theorem.
* The authors should provide source code along with easy-to-run scripts to allow reviewers to verify the reproducibility of the experiments.



[1] Universal prompt tuning for graph neural networks.

**Questions:**

See the weakness part in the review.

**Reviewer Confidence:**

2: The reviewer is willing to defend the evaluation, but it is likely that the reviewer did not understand parts of the paper

**Scope:**

3: The work is somewhat relevant to the Web and to the track, and is of narrow interest to a sub-community

---

### Official Review · Reviewer_LxQP · 2024-11-22

**Novelty:** 5
**Technical Quality:** 5

**Review:**

This paper addresses the issue of fairness awareness in pre-trained Graph Neural Networks (GNNs) and proposes the first graph prompt-based solution to reduce bias in pre-trained GNNs while enhancing adaptability across various downstream tasks. The approach employs an innovative hybrid graph prompt to augment the model with counterfactual data, alongside edge modification to enhance model heterogeneity, thus jointly improving GNN fairness.

**Advantages**:

1. The paper is well-written, clearly explaining the proposed FPrompt method and its theoretical underpinnings.

2. The paper provides a comparative analysis of FPrompt with various baseline models, demonstrating its strengths in balancing fairness and accuracy.

3. The introduction of a new benchmark for evaluating the transferability of pre-trained graph models adds value.

**Disadvantages**:
1. The paper should illustrate more about the fairness problem of pretrained models. Table 2 is not enough to demonstrate the problem well.

2. In Section 4.1.1 on hybrid graph prompting, it would be helpful to elaborate on the operation of adjusting node representations as an alternative to modifying node-sensitive attributes.

3. More about the experiment details should be explained, such as the fairness evaluation metrics.

**Typo**:

1. In Section 6.2, "Fig. 2" should be "Table 2."

**Questions:**

1. Please explain how the concept of hybrid graph hints mentioned in the article is combined with the original graph data, and their specific role and impact in the model.

2. In Section 4.2.1, the mention of sensitive group assignments could benefit from a discussion on the allocation probability for each node. Could this probability be used to address the issue of unknown node-sensitive attributes noted in Section 4.1.1?

**Reviewer Confidence:**

3: The reviewer is confident but not certain that the evaluation is correct

**Scope:**

3: The work is somewhat relevant to the Web and to the track, and is of narrow interest to a sub-community

---

### Official Review · Reviewer_izW9 · 2024-11-25

**Novelty:** 5
**Technical Quality:** 5

**Review:**

This paper introduces FPrompt, a fairness-aware graph prompt tuning method designed to simultaneously reduce the bias in pre-trained models and enhance their generalization capabilities. The method consists of two main components: hybrid graph prompts at the feature level and edge modifications at the structural level. The paper is well-written and easy to follow, with promising experimental results.

**Questions:**

- **Ablation Studies.** Fprompt comprise two key components, Hybrid graph prompts and Heterophily-enhanced edge modification. If **ablation studies** could be provided, it would enable readers to better assess the contribution of each component.
- **Pre-training Strategy.** Only one pre-training strategy Infomax is included, lacking a variety of other pre-training methods as a foundation (such as GraphCL or SimGRACE). As a result, the method's applicability and effectiveness across a broader range of pre-training scenarios are yet to be validated.
- **Unclear Hyper-parameters Analysis.** The hyper-parameters introduced by Fprompt are not well analyzed in the main body or appendix of the paper, for example, the influence of edge modification ratio $\epsilon$, besides, the sweep space is not provided, which leaves the efficiency of FPrompt somewhat obscure。

**Reviewer Confidence:**

3: The reviewer is confident but not certain that the evaluation is correct

**Scope:**

4: The work is relevant to the Web and to the track, and is of broad interest to the community

---

### Official Review · Reviewer_gLWV · 2024-11-30

**Novelty:** 5
**Technical Quality:** 4

**Review:**

This paper proposes a fairness-aware graph prompt tuning method to mitigate bias while enhancing the generalization capabilities of pre-trained GNNs. The authors also provide theoretical analysis of standard fairness metrics in the context of the pre-training and fine-tuning paradigm. Experiments across various scenarios validate the effectiveness of the proposed method.

Pros:

+ The paper studies a valuable fairness problem under the graph prompt tuning scenario, which remains under-explored by previous works.
+ The paper is well-structured and easy to follow.
+ The work is comprehensive, providing experimental results, theoretical validation, and a new benchmark for transferable evaluation.


Cons:
+ In Section 2.3, the literature review on graph fairness is insufficient, lacking related work published in 2023 and 2024. In particular, GraphPAR[1] aims at the scenario of pre-training, which is closely related to the setting of this paper. A comparative discussion between the proposed method and GraphPAR is needed.
+ The proposed method is introduced under the assumption of a binary sensitive attribute, but there exist multi-value sensitive attributes, such as race. It is necessary to provide a further description of how to apply FPrompt to multi-value sensitive attributes.
+ Lines 404-411, in the scenario of partially accessible sensitive attributes, the calculated (Equation 6) of general feature patterns for the sensitive group may shift, resulting in ineffective counterfactual data augmentation. Section 4.1.1 would be more convincing if validation of FPrompt were conducted with partially accessible sensitive attributes.
+ The core idea behind FPrompt improving fairness is data augmentation. However, Table 2 lacks comparison with similar fairness technologies, e.g., EDITS[2], Graphair[3].

Minor typos:
+ In equations 7 and 9, the subscripts of $\alpha$ and $\beta$ should be i.
+ In the experimental section, DP and EO should correspond to $\Delta_{SP}$ and $\Delta_{EO}$ in equation 3.
+ In line 723, the semicolon (“;”) is repeated.


Reference:
+ [1] Zhang Z, Zhang M, Yu Y, et al. Endowing Pre-trained Graph Models with Provable Fairness[C]//Proceedings of the ACM on Web Conference 2024. 2024: 1045-1056.
+ [2] Dong Y, Liu N, Jalaian B, et al. Edits: Modeling and mitigating data bias for graph neural networks[C]//Proceedings of the ACM web conference 2022. 2022: 1259-1269.
+ [3] Ling H, Jiang Z, Luo Y, et al. Learning fair graph representations via automated data augmentations[C]//International Conference on Learning Representations (ICLR). 2023.

**Questions:**

This method requires knowledge of sensitive attributes. Is there a solution for cases where sensitive attributes are not available? In real-world scenarios, sensitive attributes are often completely unknown due to privacy protection reasons.

**Reviewer Confidence:**

4: The reviewer is certain that the evaluation is correct and very familiar with the relevant literature

**Scope:**

3: The work is somewhat relevant to the Web and to the track, and is of narrow interest to a sub-community